# Active trachoma among children aged 1–9 years in Ethiopia: A meta-analysis from 2019 to 2024

Kibruyisfaw Weldeab Abore[1]*, Melat Tesfaye Asebot[2], Gifty Berhanemeskel Kebede[1], Robel Tibebu Kasaye[1], Asonya Abera Akuma[3], Mahlet Minwuyelet Dagne[4], Tewobesta Fesseha Tesfaye[5], Mahlet Tesfaye Abebe[6], Estifanos Bekele Fole[6]

1 Department of Ophthalmology, St. Paul's Hospital Millennium Medical College, Addis Ababa, Ethiopia, 2 Bloomberg School of Public Health, John Hopkins University, Baltimore, Maryland, United States of America, 3 College of Medicine and Health Sciences, Arbaminch University, Arbaminch, Ethiopia, 4 St. Peter's Specialized Hospital, Addis Ababa, Ethiopia, 5 Ministry of Health Ethiopia, Addis Ababa, Ethiopia, 6 Department of Medicine, Yirgalem Hospital Medical College, Yirgalem, Ethiopia

* Kibruyisfaww@gmail.com

## Abstract

### Background

Trachoma is a leading infectious cause of blindness and of significant public health concern targeted for elimination. This study aimed to systematically summarize the magnitude of active trachoma among children aged 1–9 in Ethiopia from 2019–2024.

### Methods

The review was prospectively registered on PROSPERO (Registration number: CRD42024514026). Database searches were conducted on Google Scholar, SCOPUS, PubMed, EMBASE, and Africans Journals Online (AJOL) for studies published between January 2019–31-March-2024 and with restriction to articles published only in English. Data extraction was done using a pre-prepared Excel sheet. STATA version 17 was used to perform the analysis. Heterogeneity between studies was assessed using $I^2$ statistics and Cochrane Q. Qualitative synthesis was done to summarize the studies and random effect model was used to estimate the Pooled magnitude of active trachoma with a corresponding 95% confidence interval.

### Results

A total of 17 studies with 19793 subjects were included in the meta-analysis. The pooled magnitude of active trachoma among children aged 1–9 years was found to be 18.4% (95% CI: 13.88, 22.91). We found a statistically significant heterogeneity between studies. Among the regions, Southwest region was found to have the highest magnitude (44.1%) (95%CI: 41.8%, 46.4%) and Dire Dawa was found to have the lowest (4.3%) (95%CI: 2.9%, 5.7%).

**Data availability statement:** All relevant data are within the manuscript and its Supporting information files.

**Funding:** The author(s) received no specific funding for this work.

**Competing interests:** The authors have declared that no competing interests exist.

## Conclusion

The magnitude of active trachoma is still higher than the World Health Organization (WHO) target for elimination. There was significant interregional difference in magnitude of active trachoma. Strengthening surgical treatment for trichiasis, antibiotic therapy, facial hygiene, and environmental improvement (SAFE) strategy and health education and promotion is recommended.

## Introduction

Trachoma is an infectious ocular disease caused by the bacteria Chlamydia trachomatis and it is the leading infectious cause of blindness worldwide [1]. Active trachoma is defined by the presence of trachomatous inflammation–follicular (TF) and/or trachomatous inflammation–intense (TI) in at least one eye [2]. Trachoma primarily affects impoverished communities within low-income and middle-income nations. Trachoma is endemic in 57 countries globally, and it is responsible for the blindness and visual impairment of nearly 2 million people [3–5].

Africa bears the largest burden of trachoma globally with prevalence of TF ≥30% at most recent surveys [6]. Trachoma is the second most common cause of blindness in Ethiopia and still constitutes a disease of public health concern [7].

Trachoma disproportionately affects disadvantaged and marginalized groups within communities [8]. In endemic areas, active cases of the disease are common among the preschool children, with varying magnitude [9]. Children exhibiting the active stages of trachoma also serves as the reservoir for the transmission of infection [10]. Trachoma is managed through an endorsed, comprehensive strategy comprising of surgical treatment for trichiasis, antibiotic therapy, facial hygiene, and environmental improvement, collectively known as the SAFE strategy [11].

The relevance of this study is multifaceted. Firstly, it updates and expands upon previous research by including data from the past five years, a period marked by significant socio-political upheavals in Ethiopia. A previous meta-analysis done in Ethiopia in 2019 had shown the magnitude of active trachoma among children to be 26.9% [8]. This magnitude is expected to rise due to the wars, conflicts, and its associated internal displacements in Ethiopia over the past 5 years. Furthermore, public health efforts would be affected which would negatively affect the progress towards trachoma elimination [12]. In addition, multiple studies were carried out in various regions of Ethiopia over the last five years and significant discrepancies in magnitude were seen. This concern highlights the need to combine the findings from these studies to establish evidence-based suggestions. Thus, this study aimed to systematically summarize and pool the magnitude of active trachoma among children aged 1–9 years in Ethiopia from 2019–2024.

## Method and materials

### Study design

This systematic review and meta-analysis was prospectively registered on PROSPERO (Registration number: CRD42024514026).

## Protocol amendment

There were few deviations from the initial protocol registered and updated protocol is available on PROSPERO.

## Search strategy and sources

Two reviewers independently conducted searches on Google Scholar, SCOPUS, PubMed, EMBASE, and AJOL. The search was made using the syntax (((("Active trachoma"[Title/Abstract]) OR (trachoma[Title/Abstract])) AND (school children[Title/Abstract])) OR (children[Title/Abstract])) AND (Ethiopia[Title/Abstract]). The search was restricted to articles published between January 2019 to March 31 2024 and to those published in English. Literatures were rechecked after completion of the study. Preferred Reporting Items for Systematic reviews and Meta-Analyses (PRISMA) 2020 guideline was used to report the study selection steps.

## Eligibility criteria

The inclusion criteria for the study was studies that were conducted between 2019–2024, that are published in English, and that assessed the magnitude of active trachoma among children 1–9 years in Ethiopia. Studies that did not report on the outcome of interest, studies done before 2019, those conducted outside of Ethiopia, studies which are not published in English language, and studies without retrievable full article were excluded from the study.

## Quality appraisal

The Newcastle-Ottawa scale (NOS) adapted for cross-sectional studies was used to assess the quality of studies included in the review [13]. It assess selection, comparability, and outcome domains of studies with 5, 2, and 3 stars maximum given for selection, comparability, and outcome domains respectively. Two reviewers independently assessed the quality of studies. Studies with a score of less than 7 were labeled as poor and those with ≥7 were assessed as good quality studies. A third reviewer settled disagreement in scoring between the two reviewers through discussions.

## Data extraction

Identified literature were imported to Covidence and examined against the eligibility criteria by two reviewers. A third reviewer served as a mediator when there was disagreement. Using a pre-prepared excel sheet, two reviewers with the help of a third reviewer, extracted data. We collected data regarding the name of the primary author, publication year, year of study conduct, region of the study, study design, sample size, and active trachoma magnitude with 95% confidence interval. Missing data on confidence interval of studies were handled through communicating the authors of the studies and through imputations.

## Data processing and analysis

The extracted data were exported to STATA v.17 for analysis. Cochran's Q test and $I^2$ statistics were used to assess heterogeneity among studies. Using a random effects model, we computed the pooled magnitude of active trachoma among children and presented it using a forest plot. A Subgroup analysis was conducted based on region and Meta-regression was done using sample size and year of publication as covariates. A leave one out sensitivity analysis was computed. We assessed publication bias using Funnel plot and Eggers test.

# Results

## Study screening and selection

The search strategy identified a total of 570 studies. Subsequently, 98 duplicate studies were removed. After reviewing title and abstract, 422 studies were removed. Full article screening was done on 50 articles, and 9 studies that did not

report on magnitude of trachoma, 21 studies based on year of study, 1 study done on adult subjects, and 2 studies without clear study period were excluded. Finally, 17 studies were found to be eligible for the meta-analysis (Fig 1).

## Study characteristics

Of the studies, 5 were done in Amhara region [14–18], 3 in Oromia region [19–21], 2 from Southen region [22,23], 3 in Gambella [24–26], 1 from South West Ethiopia region[27], 1 in Benishangul Gumuz [28], 1 in Sidama [29], and 1 in Dire-dawa city [30]. Based on NOS assessment of quality, all included studies had good quality (a score ≥7). In terms of the number of study participants, the largest sample size was 5489 [25] and smallest was 178 [20]. All included studies were cross-sectional study design (Table 1).

## Meta-analysis of Active trachoma among children 1–9 years of age in Ethiopia

We included 17 studies for the meta-analysis with a total sample size of 19793. Due to the statistically significant heterogeneity among studies, we used random effect model to estimate the pooled magnitude of active trachoma. The

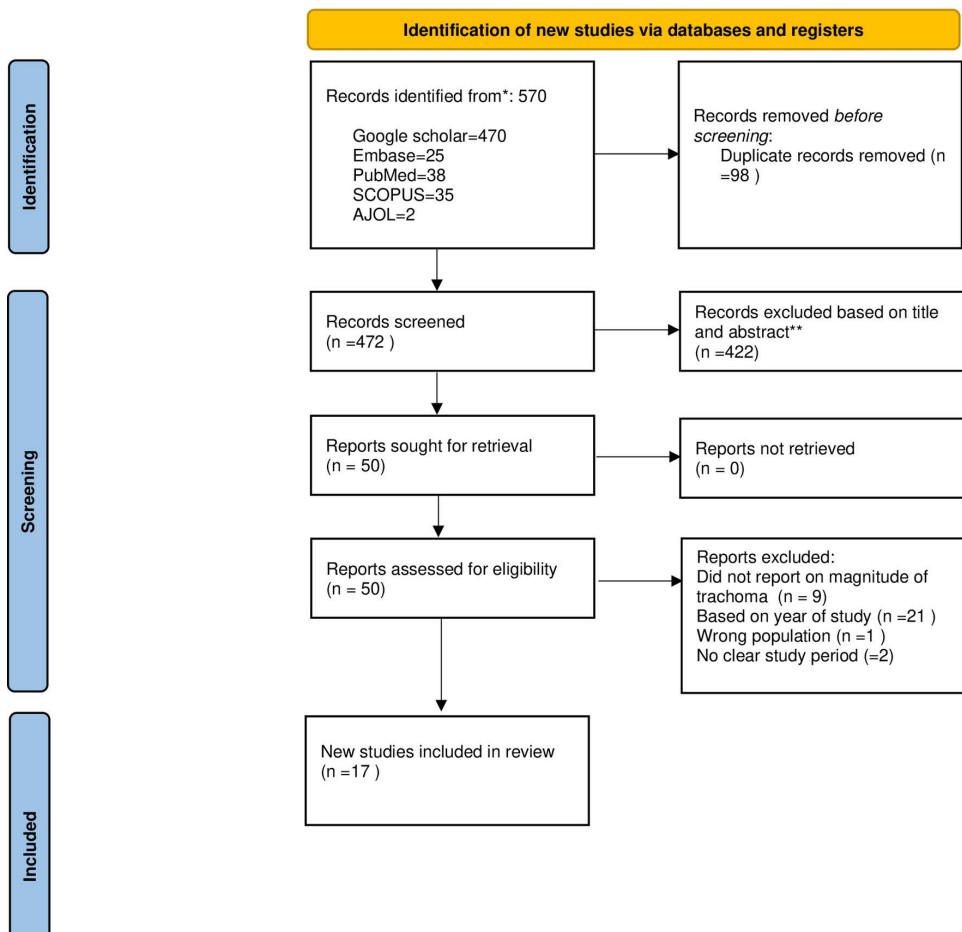

**Fig 1. PRISMA study selection flow for systematic review and meta-analysis of active trachoma among children aged 1–9 years in Ethiopia: A meta-analysis from 2019 to 2024.**

magnitude of active trachoma among children was found to be 18.4% (95% CI: 13.88, 22.91) (Fig 2). Meta-regression done using sample size and year of study found that both are not statistically significant sources of heterogeneity (Table 2). Subgroup analysis based on the region of the study was conducted and between studies heterogeneity persisted. Southwest region was found to have the highest magnitude of active trachoma (44.1%) while Dire Dawa City was found to have the lowest magnitude (4.3%) (Table 3).

### Sensitivity analysis

A leave-one-out sensitivity analysis was conducted to assess the effect of one study on the overall effect size estimate. We found that no single study had a significant effect on the overall effect size estimate (Table 4).

### Publication bias

We assessed publication bias using a funnel plot and Egger's test. The funnel plot was found to be slightly asymmetric to the right (Fig 3) and Egger's test showed a statistically significant publication bias (p-value = 0.0112). Subsequently, trim and fill analysis was done with no change in the estimated effect size. The result of Egger's test was assumed to be originating from the statistically significant heterogeneity.

## Discussion

The study showed that there is a high prevalence of active trachoma among children 1–9 years in Ethiopia, with significant variation in prevalence among different regions of the country. The pooled variance in this study is 18.4% which is markedly higher than the World Health Organization's (WHO) target of trachoma elimination of less than 5% active trachoma [4]. The finding is comparatively lower than previous report by Gebrie et al in 2019 that showed a pooled prevalence of

**Table 1. Characteristics of studies included for systematic review and meta-analysis on active trachoma among children aged 1–9 years in Ethiopia from 2019-2024.**

| Authors | Year of publication | Region | Study design | SS | NOS score |
|---|---|---|---|---|---|
| Asmare Z et al. [16] | 2023 | Amhara | Cross-sectional | 540 | 8 |
| Altaseb et al. [15] | 2024 | Amhara | Cross-sectional | 616 | 9 |
| Genet et al. [14] | 2022 | Amhara | Cross-sectional | 704 | 9 |
| Melkie et al. [17] | 2020 | Amhara | Cross-sectional | 678 | 9 |
| Shimelash et al. [18] | 2022 | Amhara | Cross-sectional | 394 | 9 |
| Mekonnen et al. [20] | 2022 | Oromia | Cross-sectional | 178 | 7 |
| Shafi et al. [21] | 2023 | Oromia | Cross-sectional | 526 | 7 |
| Tuke et al. [19] | 2023 | Oromia | Cross-sectional | 538 | 8 |
| Mengiste et al. [28] | 2022 | Benishangul | Cross-sectional | 4590 | 7 |
| Mohamed et al. [30] | 2019 | Diredawa | Cross-sectional | 823 | 8 |
| Delelegn et al. [29] | 2021 | Sidama | Cross-sectional | 701 | 7 |
| Getachew et al. [27] | 2023 | South West Ethiopia | Cross-sectional | 1292 | 10 |
| Alemayehu et al. [25] | 2023 | Gambella | Cross-sectional | 5489 | 7 |
| Senebete et al. [24] | 2024 | Gambella | Cross-sectional | 722 | 8 |
| Yitayeh et al. [26] | 2021 | Gambella | Cross-sectional | 610 | 8 |
| Abdilwohab et al. [22] | 2020 | Southern region | Cross-sectional | 831 | 9 |
| Shemsu et al. [23] | 2021 | Southern region | Cross-sectional | 561 | 8 |

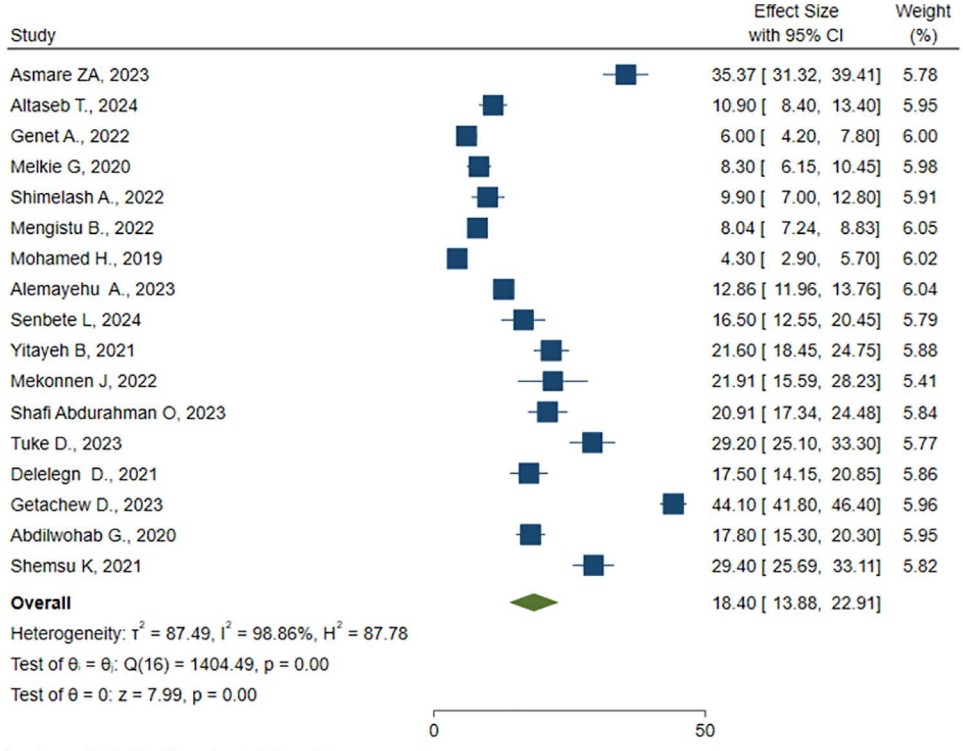

**Fig 2. Pooled magnitude of active trachoma among children aged 1–9 years in Ethiopia from 2019-2024.**

**Table 2. Meta-regression analysis of factors affecting between-study heterogeneity.**

| Source | Coefficient | Std. error | p-value |
|---|---|---|---|
| Year of publication | 2.81268 | 1.87961 | 0.135 |
| Sample size | -0.02064 | 0.00180 | 0.253 |

**Table 3. Subgroup analysis based on region for active trachoma among children aged 1–9 years in Ethiopia from 2019-2024.**

| Region | Pooled magnitude | 95% CI | Heterogeneity ($I^2$) |
|---|---|---|---|
| Amhara | 14% | 3.57, 24.44 | 98.87% |
| Oromia | 24.11% | 18.71, 29.5 | 76.59% |
| Southern region | 23.51% | 12.14, 34.88 | 96.13% |
| Sidama | 17.5% | 14.15, 20.85 | – |
| Gambella | 16.82% | 11.68, 21.96 | 91.43% |
| Southwest region | 44.1% | 41.8, 46.4 | – |
| Dire Dawa | 4.3% | 2.9, 5.7 | – |
| Benishangul Gumuz | 8.04% | 7.24, 8.83 | – |

26.9% and similar disparities among regions. [8] Although their study reported a similar pattern and trend, the study period was limited to the year 2019. Our study extended from 2019–2024 which makes it relevant in capturing effects of recent occurrences such as the ongoing war and conflicts in the country over the past five years.

Previous studies have highlighted the significant consequence of war and displacement in hindering disease control by disrupting healthcare infrastructure, reducing access to clean water and sanitation, and spreading infectious diseases because of migration [5,12]. Despite the current ongoing war and displacement in the country, the pooled prevalence of our study is lower than the one reported 5 years ago (26.9% Vs 18.4%). One possible reason could be that data from Tigray and Afar, regions which are heavily affected by the war, are not included in our study due to a lack of data from those regions. Therefore, the reported prevalence might underestimate the true prevalence.

**Table 4. A leave one out sensitivity analysis for studies on active trachoma among children aged 1-9 years in Ethiopia from 2019-2024.**

| Omitted study | Effect size | [95% conf. interval] | | p-value |
|---|---|---|---|---|
| | | Upper | Lower | |
| Asmare ZA, 2023 | 17.348 | 12.873 | 21.824 | 0.000 |
| Altaseb T., 2024 | 18.875 | 14.118 | 23.632 | 0.000 |
| Genet A., 2022 | 19.192 | 14.418 | 23.967 | 0.000 |
| Melkie G, 2020 | 19.043 | 14.269 | 23.817 | 0.000 |
| Shimelash A., 2022 | 18.933 | 14.207 | 23.660 | 0.000 |
| Mengistu B., 2022 | 19.084 | 13.824 | 24.343 | 0.000 |
| Mohamed H., 2019 | 19.303 | 14.556 | 24.051 | 0.000 |
| Alemayehu A., 2023 | 18.781 | 13.338 | 24.224 | 0.000 |
| Senbete L, 2024 | 18.515 | 13.830 | 23.199 | 0.000 |
| Yitayeh B, 2021 | 18.197 | 13.536 | 22.857 | 0.000 |
| Mekonnen J, 2022 | 18.195 | 13.553 | 22.838 | 0.000 |
| Shafi Abdurahman O, 2023 | 18.241 | 13.577 | 22.905 | 0.000 |
| Tuke D., 2023 | 17.732 | 13.154 | 22.309 | 0.000 |
| Delelegn D., 2021 | 18.454 | 13.760 | 23.148 | 0.000 |
| Getachew D., 2023 | 16.623 | 13.369 | 19.877 | 0.000 |
| Abdilwohab G., 2020 | 18.438 | 13.712 | 23.163 | 0.000 |
| Shemsu K, 2021 | 17.712 | 13.153 | 22.271 | 0.000 |
| Theta | 18.396 | 13.884 | 22.908 | 0.000 |

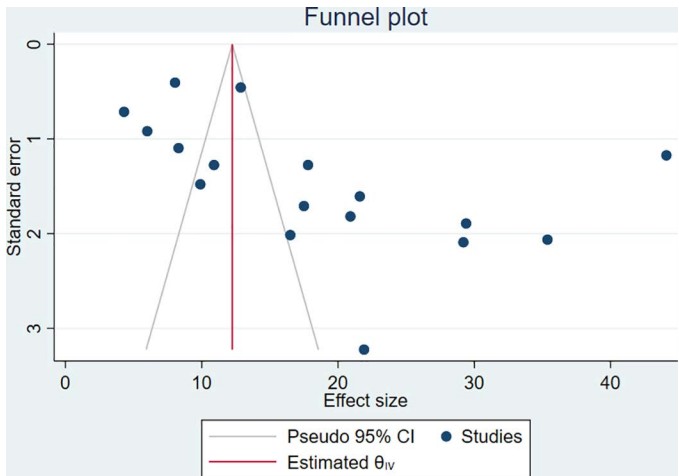

**Fig 3. Funnel plot assessing publication bias for active trachoma among children aged 1-9 years in Ethiopia from 2019-2024.**

Several other factors may contribute to this high prevalence rate that persisted in the country despite implementing control measures to reduce the burden. The low socioeconomic status of the country is the major one which results in a lack of access to clean water and inadequate sanitation. According to UNICEF and WHO's report, 48% of the population lacks access to clean water and functional latrines, which significantly impacts hygiene practices and increases the risk of trachoma transmission [31]. Lack of hygiene is a significant factor in the spread of infection. Global studies have indicated that countries with poor sanitation and water supply have higher trachoma prevalence, underscoring the need for comprehensive public health interventions [32–34]. A study by Garn et al. demonstrated the inverse relation between adequate water and sanitation facilities and trachoma prevalence [2]. Poverty is also inversely related to education and health care access which are essential factors to the successful implementation of control measures. Trachoma was indicated as an indicator of poverty [35].

Cultural practices are another factor contributing to the high prevalence of trachoma. Some practices such as using clothes to clean eyes, inadequate showers, and facial cleanliness further been reported to increase the risk of trachoma transmission [36]. War and conflict on top of this lays a fertile ground for the increase in transmission.

To reduce Trachoma prevalence the WHO recommended SAFE (Surgery for trichiasis, Antibiotic to clear infection, Facial cleanliness, and Environmental improvement) strategy should be strengthened [6,37]. Ethiopia as a member of WHO also started to implement SAFE strategy since 2003 to tackle the high burden of trachoma [38,39]. Building infrastructures for Water, Sanitation, and Hygiene (WASH), with special emphasis to high prevalence area, is critical to hinder the transmission of trachoma through addressing the facial cleanliness and environmental improvement component of SAFE strategy [3,6].This finding is shown by a meta-analysis conducted in Ethiopia in 2021 that showed households without access to toilet facilities, lacking access improved water, and not practicing regular face washing for children had higher likelihood of having active trachoma [40]. Empowering the public by raising awareness about the severity of the prevalence and the resulting blindness as well as the hygiene techniques could ease the burden and should be widely implemented. A due focus also should be given to expanding and equipping healthcare facilities with trained staff and resources to combat trachoma. Lastly, a robust surveillance system to monitor the prevalence and impact of intervention should be in play to monitor and evaluate the implementations.

Eleven countries, including 3 from Africa (The Gambia, Ghana, Morocco), have eliminated trachoma in 2022 [1]. This could indicate public health interventions such as the sustained SAFE strategy implementation can lead to the elimination of trachoma. These countries serve as a great example and those strategies should be tailored to the country's status and implemented to achieve the elimination target.

## Limitation

Despite its strength in shedding light on the prevalence of trachoma, our study has some limitations. The first is that the included studies are cross-sectional studies hence limiting causal inferences. Second, the exclusion of conflict-affected regions like Tigray and Afar due to a lack of recent data may result in an underestimation of the true prevalence of active trachoma in Ethiopia. This warrants cautious interpretation of the result. Although extensive attempts were made to identify studies that reported prevalence of trachoma, selection bias could have been introduced due to missed studies.

## Conclusion

Although it has decreased from the previously reported magnitude, active trachoma is still higher than the WHO target for elimination and is a disease of public health concern. We recommend strengthening the SAFE strategy to better tackle the problem as well as working with relevant stakeholders about health education and health promotion. Future research should focus on longitudinal analysis to better understand the transmission of trachoma and associated socio-demographic factors in different regions of the country. The long-term impact of the ongoing intervention should also be

assessed in future studies. These studies will help tailor public health interventions based on the specific findings in the regions. Additionally, research comparing trachoma control efforts in Ethiopia with those in other high-prevalence countries can provide valuable insights into effective strategies and potential areas for improvement.

## Supporting information

**S1 File. PRISMA 2020 checklist.**
(DOCX)

**S2 File. Data extraction summary sheet.**
(XLSX)

**S3 File. All studies identified.**
(DOCX)

**S4 File. Newcastle Ottawa scale quality assessment.**
(DOCX)

## Author contributions

**Conceptualization:** Kibruyisfaw Weldeab Abore.

**Data curation:** Kibruyisfaw Weldeab Abore, Robel Tibebu Kasaye, Asonya Abera Akuma, Mahlet Minwuyelet Dagne, Tewobesta Fesseha Tesfaye.

**Formal analysis:** Kibruyisfaw Weldeab Abore, Melat Tesfaye Asebot, Robel Tibebu Kasaye, Asonya Abera Akuma, Mahlet Minwuyelet Dagne, Tewobesta Fesseha Tesfaye.

**Investigation:** Kibruyisfaw Weldeab Abore, Melat Tesfaye Asebot, Gifty Berhanemeskel Kebede, Robel Tibebu Kasaye, Mahlet Minwuyelet Dagne, Tewobesta Fesseha Tesfaye, Mahlet Tesfaye Abebe, Estifanos Bekele Fole.

**Methodology:** Kibruyisfaw Weldeab Abore, Melat Tesfaye Asebot, Gifty Berhanemeskel Kebede, Asonya Abera Akuma, Mahlet Minwuyelet Dagne, Mahlet Tesfaye Abebe.

**Software:** Kibruyisfaw Weldeab Abore, Melat Tesfaye Asebot, Gifty Berhanemeskel Kebede, Asonya Abera Akuma, Mahlet Minwuyelet Dagne, Tewobesta Fesseha Tesfaye, Estifanos Bekele Fole.

**Supervision:** Robel Tibebu Kasaye, Tewobesta Fesseha Tesfaye.

**Validation:** Gifty Berhanemeskel Kebede, Robel Tibebu Kasaye, Asonya Abera Akuma, Tewobesta Fesseha Tesfaye, Mahlet Tesfaye Abebe, Estifanos Bekele Fole.

**Visualization:** Estifanos Bekele Fole.

**Writing – original draft:** Kibruyisfaw Weldeab Abore, Melat Tesfaye Asebot, Gifty Berhanemeskel Kebede, Asonya Abera Akuma, Mahlet Tesfaye Abebe.

**Writing – review & editing:** Kibruyisfaw Weldeab Abore, Melat Tesfaye Asebot, Gifty Berhanemeskel Kebede, Robel Tibebu Kasaye, Mahlet Minwuyelet Dagne, Tewobesta Fesseha Tesfaye, Mahlet Tesfaye Abebe, Estifanos Bekele Fole.

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
