## [Decision Letter · Decision Letter 0]

9 Feb 2025

PONE-D-24-40061Active trachoma among children aged 1-9 years in Ethiopia: A meta-analysis from 2019 to 2024PLOS ONE

Dear Dr. Abore,

Thank you for submitting your manuscript to PLOS ONE. After careful consideration, we feel that it has merit but does not fully meet PLOS ONE’s publication criteria as it currently stands. Therefore, we invite you to submit a revised version of the manuscript that addresses the points raised during the review process.

We look forward to receiving your revised manuscript.

Kind regards,

Dawit Getachew

Gust Academic Editor

PLOS ONE

Journal Requirements:

2. As required by our policy on Data Availability, please ensure your manuscript or supplementary information includes the following:

We appreciate the authors for the excellent work done. The meta-analysis highlights the public health problem and is highly relevant with detailed eligibility criteria.

I request Authors to address the following queries!

1, Why non-peer reviewed studies were excluded?

2, While comparing to previous studies, consider discussing any methodological differences that might account for discrepancies.Al so further discussion on confounding variables would strengthen.

Additional Editor Comments:

Dear author, Thank you for submitting your manuscript titled "Active trachoma among children aged 1-9 years in Ethiopia: A meta-analysis from 2019 to 2024" to our journal. After careful consideration, we have found the manuscript can be published in this journal, but it is not in its current form. We have received the required number of review reports. Based on the reviewer comment and suggestion, we have decided the manuscript needs major revision. Therefore, in the revised manuscript,, try to address points raised by reviwewrs.

Reviewers' comments:

Reviewer's Responses to Questions

**Comments to the Author**

1. Is the manuscript technically sound, and do the data support the conclusions?

Reviewer #1: Yes

Reviewer #2: Yes

2. Has the statistical analysis been performed appropriately and rigorously? 

Reviewer #1: Yes

Reviewer #2: Yes

3. Have the authors made all data underlying the findings in their manuscript fully available?

Reviewer #1: Yes

Reviewer #2: Yes

4. Is the manuscript presented in an intelligible fashion and written in standard English?

Reviewer #1: Yes

Reviewer #2: Yes

5. Review Comments to the Author

Reviewer #1: Very good meta-analysis. I would recommend adding more on the possible ways to tackle the trachoma health crisis in certain countries like analyzing the SAFE techniques more for the better understanding of the reader.

Reviewer #2: First of all, I would like to thank the authors for their efforts. This manuscript is titled Active trachoma among children aged 1-9 years in Ethiopia: A meta-analysis from 2019 to 2024. Although the overall quality of this manuscript is good, the following comments are for improving its quality:

1. In the Methods section, please submit your full search syntax for each database as a supplementary file.

2. Which version of PRISMA was used?

3. In Figure 1, please specify how many studies each database had.

4. Given the local nature of your topic, it is recommended to include studies published in your local language. This will reduce selection bias.

5. How did you deal with grey literature? Because according to PRISMA and Cochrane, it is recommended to include grey literature studies in the search and perform sensitivity analysis on them.

6. Is there no IRB code for this manuscript?

7. Are there any deviations from the protocol registered in Prospero? If so, please write under the title protocol amendment.

8. All abbreviations should be written in full the first time they are used.

9. Mention any limitations of your study, including selection bias, etc.

6. PLOS authors have the option to publish the peer review history of their article (what does this mean? ). If published, this will include your full peer review and any attached files.

**Do you want your identity to be public for this peer review?** For information about this choice, including consent withdrawal, please see our Privacy Policy .

Reviewer #1: No

Reviewer #2: No

---

## [Author Response · Author response to Decision Letter 1]

3 Apr 2025

Dear editor,

We would like to take a moment to thank the Editor and the peer reviewers for the constructive evaluation of our paper. We have reviewed the comments provided by the editor and the reviewers and have carefully edited the manuscript. We have also provided responses accordingly.

Sincerely,

Kibruyisfaw Weldeab Abore

Editorial comments

1. A numbered table of all studies identified in the literature search, including those that were excluded from the analyses. For every excluded study, the table should list the reason(s) for exclusion. If any of the included studies are unpublished, include a link (URL) to the primary source or detailed information about how the content can be accessed.

Response: thank you for the comment. We have included a table of all identified studies as a supplementary file

2. A table of all data extracted from the primary research sources for the systematic review and/or meta-analysis. The table must include the following information for each study: Name of data extractors and date of data extraction, Confirmation that the study was eligible to be included in the review. All data extracted from each study for the reported systematic review and/or meta-analysis that would be needed to replicate your analyses. If data or supporting information were obtained from another source (e.g. correspondence with the author of the original research article), please provide the source of data and dates on which the data/information were obtained by your research group.

Response: thank you for the comment. We have included a table of all data extracted as a supplementary file

3. If applicable for your analysis, a table showing the completed risk of bias and quality/certainty assessments for each study or outcome. Please ensure this is provided for each domain or parameter assessed. For example, if you used the Cochrane risk-of-bias tool for randomized trials, provide answers to each of the signaling questions for each study. If you used GRADE to assess certainty of evidence, provide judgments about each of the quality of evidence factor. This should be provided for each outcome.

Response: thank you for the comment. We have included a table showing ROB assessment for each studies.

Response: thank you for the comment. We have included a statement on how missing data was handled.

5. Why non-peer reviewed studies were excluded?

Response: thank you for the comment. We put this exclusion criteria initially while preparing the protocol. We understand that it can’t introduce selection bias. However, no study that fulfilled the eligibility criteria was excluded due to this criteria. We have removed it from the eligibility criteria.

6. While comparing to previous studies, consider discussing any methodological differences that might account for discrepancies. Also further discussion on confounding variables would strengthen.

Response: Thank you for the comment. We accept the comment and have edited the section.

Reviewer 1

1. Very good meta-analysis. I would recommend adding more on the possible ways to tackle the trachoma health crisis in certain countries like analyzing the SAFE techniques more for the better understanding of the reader.

Response: Thank you for the comment. We have edited the manuscript based on the insight provided.

Reviewer 2

1. In the Methods section, please submit your full search syntax for each database.

Response: Thank you for the comment. We have included the search used to identify literatures from databases

2. Which version of PRISMA was used?

Response: thank you for the comment. We have included the version of PRISMA in the manuscript.

3. In Figure 1, please specify how many studies each database had.

Response: Thank you for the comment. We have specified the number of studies identified

4. Given the local nature of your topic, it is recommended to include studies published in your local language. This will reduce selection bias.

Response: Thank you for the comment. We acknowledge including those published in English language only would introduce selection bias. However, we did not identify any article that assessed the outcome of interest published in local languages.

5. How did you deal with grey literature? Because according to PRISMA and Cochrane, it is recommended to include grey literature studies in the search and perform sensitivity analysis on them.

Response: thank you for the comment. The search was made on prelisted databases. Grey literatures including those from university repositories were made. Manual search of literatures was done from list of references of articles as well.

6. Is there no IRB code for this manuscript?

Response: thank you for the comment. Since no data was collected from human subject for this study, Ethical approval is waived.

7. Are there any deviations from the protocol registered in Prospero? If so, please write under the title protocol amendment.

Response: thank you for the comment. We have edited the manuscript based on comments.

8. All abbreviations should be written in full the first time they are used.

Response: thank you for the comment. We have amended the manuscript based on the comment.

9. Mention any limitations of your study

Response: Thank you for the comment. We have added the limitations of the study.

---

## [Editor Report · Decision Letter 1]

11 Apr 2025

Active trachoma among children aged 1-9 years in Ethiopia: A meta-analysis from 2019 to 2024

PONE-D-24-40061R1

Dear Dr. Kibruyisfaw,

We’re pleased to inform you that your manuscript has been judged scientifically suitable for publication and will be formally accepted for publication once it meets all outstanding technical requirements.

Kind regards,

Dawit Getachew Gebeyehu, MPH

Academic Editor

PLOS ONE
---

## [Editor Report · Acceptance letter]

PONE-D-24-40061R1

PLOS ONE

Dear Dr. Abore,

I'm pleased to inform you that your manuscript has been deemed suitable for publication in PLOS ONE. Congratulations! Your manuscript is now being handed over to our production team.

Kind regards,

on behalf of

Mr. Dawit Getachew Gebeyehu

Academic Editor

PLOS ONE